# GoalAct: A Globally Adaptive Dynamic Legal Multi-agent Collaboration System

**Junjie Chen**
2024210901

**Ruowen Zhao**
2024210905

**Zhiyuan Feng**
2024311588

## Abstract

This paper presents our proposed multi-agent collaboration system based on GLM-4, which employs a strategy that combines global and local information to provide legal services by accessing relevant legal databases through API calls. The strength of our approach lies in integrating planning, reflection and memory globally and locally, thereby enhancing both the accuracy and adaptability.

## 1 Background

Recently, Large Language Models (LLMs) have emerged as a major advancement in artificial intelligence and natural language processing. With advanced natural language understanding and generation capabilities, LLMs have moved beyond simple text generation and question-answering tasks and become a powerful foundation for developing intelligent agents. For instance, when integrated as legal agents, LLMs can handle personal inquiries and help search for case-related information by accessing external legal databases through API calls. Such legal agents can improve the efficiency of services and simplify the form of legal consultation, helping to address the limited availability and high cost of legal professionals, especially in regions where legal services are often restricted [1].

Despite those advantages, developing an efficient legal multi-agent system based on LLMs still presents a number of challenges. First, due to the complexity and diversity of user inquiries, the system must be able to effectively filter out irrelevant or redundant information when processing user inputs to ensure accurate responses. Second, to handle a wide variety of legal issues, the system should be capable of generating logical and coherent planning paths and avoid being trapped in local search loops, which may leads to no responses. It is also crucial for the system to develop a robust self-correction mechanism to improve its reliability. Moreover, the system also needs to form memory and accumulate prior experience to reduce repeated errors, allowing for enhanced decision-making abilities over time.

In this project, by utilizing the open-source advanced language model GLM-4 [2], we have developed a globally adaptive dynamic legal multi-agent collaboration system by accessing databases through APIs to address the aforementioned challenges. Our experimental results also demonstrate its superior performance for legal services.

## 2 Definition

Specifically, given user demands $U$, we aim to produce a solution $S$ by accessing legal databases through APIs. The process can be represented as:

$$S = MultiAgent(U; APIs)$$

Our goal is to minimize the discrepancy between the solution $S$ and the user demands $U$:

38th Conference on Neural Information Processing Systems (NeurIPS 2024).

$$\min_{S} \mathcal{L}(S, U)$$

where $\mathcal{L}$ is a loss function measuring the difference between $S$ and $U$, ensuring that each agent's local task aligns with the global objective.

# 3 Related Work

Recent advancements in LLMs have led to significant progress in enabling models to perform complex reasoning tasks. The **Chain-of-Thought** (CoT) [3] approach allows models to generate intermediate reasoning steps, effectively breaking down complex problems into simpler sub-tasks. This methodology enhances the model's ability to handle tasks that require multi-step reasoning by making the thought process explicit.

**Basic reflection** [4] mechanisms have been introduced to enable models to assess and refine their own outputs. By reflecting on previous responses, models can identify errors or inconsistencies and make necessary adjustments, leading to improved performance and reliability.

**ReAct** [5] framework combines reasoning and acting within language models, allowing agents to plan actions and update their knowledge base dynamically. This synergistic approach enables models to interact with the environment and adapt their strategies based on new information.

**Reflexion** [6] introduces language agents with verbal reinforcement learning capabilities, enabling them to learn from feedback and refine their decision-making processes. By incorporating reinforcement signals, these agents can adjust their actions to achieve better outcomes over time.

# 4 Proposed Method

We propose a globally adaptive dynamic legal multi-agent collaboration system composed of five types of agents: **Processor**, **Memorizer**, **Actor**, **Judge**, and **Reflector**. When user demands are input into the system, the **Processor** corrects errors and duplicates in the queries and filters the external information needed to address the problem. Subsequently, the **Memorizer** distills short-term memories to aid information synchronization among the subsequent agents and retrieves long-term memories related to various issues, thereby enhancing the agents' problem-solving capabilities. The **Actor** agent, integrating the think-act-observe components, devises and executes solution plans based on relevant memories. These plans are dynamically adjusted in response to feedback from the external environment, ultimately leading to a resolution of the problem. The **Judge** then assesses the quality of the solution, outputting it to the user if it completely resolves the issue. If the solution does not fully meet the user's needs, the **Reflector** first refines and summarizes the existing information, then reviews the Actor's plans, and formulates an improved solution.

In our approach, each agent is assigned subtasks within their capability, significantly reducing the complexity of problem-solving and enhancing the system's overall effectiveness. Although the multi-agent collaboration paradigm effectively enhances performance by decomposing complex problems into manageable subproblems, if each agent focuses solely on their tasks without considering the overall objective, it can degrade the system's performance. Thus, a superior multi-agent system must not only enhance the accuracy of local tasks but also ensure alignment with the global objective. **Balancing the accuracy of local tasks with the consistency of global objectives poses a critical challenge for multi-agent collaboration systems.**

To address this challenge, we innovatively propose **GoalAct**, which considers local accuracy and global consistency across planning, reflection, and memory dimensions. As for planning, the system considers the overall goal during local planning and adjusts the global plan based on dynamic feedback from local planning to enhance the overall effect. As for reflection, the system reflects in real-time on its actions during interactions with the external world and also reflects on the overall plan to determine if the current planning direction meets the overall objectives, thereby enhancing reliability. As for memory, the system uses short-term memory to summarize useful information from one agent and passes it to the next, facilitating more efficient task completion. It also leverages long-term memory to continuously accumulate past experiences and lessons, thereby enhancing its capability to handle different task types and improving its generalizability.

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
