# OpenReview forum: "【Proposal】GoalAct: A Globally Adaptive Dynamic Legal Multi-agent Collaboration System"
_tsinghua.edu.cn/THU/2024/Fall/AML — THU 2024 Fall AML Submission_

### Official Review · ~王俊逸1 · 2024-11-08
**Innovative Approach to Legal Services through Multi-Agent Collaboration**

**Rating:** 8
**Confidence:** 4

**Review:**

The proposal for "GoalAct: A Globally Adaptive Dynamic Legal Multi-agent Collaboration System" presents a forward-thinking approach to leveraging Large Language Models (LLMs) for legal services. It addresses the complexity of user inquiries and the need for a robust, self-correcting system capable of accessing legal databases through API calls. The integration of planning, reflection, and memory at both global and local levels is a significant strength, enhancing the system's accuracy and adaptability.

The proposal's methodology, which includes a diverse set of agents—Processor, Memorizer, Actor, Judge, and Reflector—shows promise in streamlining legal processes and improving decision-making. The system's ability to dynamically adjust plans based on feedback and to learn from past experiences is particularly innovative, potentially leading to more efficient and effective legal services.

However, the proposal could benefit from a more detailed discussion on how the system will handle the confidentiality and sensitivity of legal data, as well as the ethical implications of automating legal advice. Additionally, while the system's adaptability is a key feature, it is crucial to ensure that the system's global objectives do not compromise the autonomy and creativity of individual agents.

Overall, the proposal for GoalAct is a commendable effort to revolutionize the legal sector through advanced AI technology. It presents a comprehensive system that could significantly improve the accessibility and quality of legal services, provided that the implementation addresses the associated ethical and data security concerns.

---

### Official Review · ~Xiying_Huang2 · 2024-11-09
**Evaluation of “GoalAct: A Globally Adaptive Dynamic Legal Multi-agent Collaboration System”**

**Rating:** 8
**Confidence:** 3

**Review:**

The quality of the paper is commendable, as it addresses a complex problem in legal multi-agent collaboration with a well-structured approach that integrates advanced AI methodologies such as planning, memory, and reflection within the GLM-4 framework. The paper is logically organized, with each section providing clarity on the model’s components and their roles within the system.

The clarity of the paper is generally high, though a few technical aspects could benefit from additional explanation. Terms like “dynamic feedback from local planning” and the specific functionalities of agents (e.g., “Memorizer” and “Reflector”) are well introduced, but some readers may find these ideas abstract without further clarification. Including illustrative examples could enhance readability and help audiences understand the practical application of each component.

The paper shows originality by developing a multi-agent legal system that combines local and global strategies to manage complex legal inquiries. Leveraging GLM-4 for such applications demonstrates a novel approach, especially through the use of reinforcement mechanisms and multi-agent collaboration for legal services.

This work is significant as it targets the under-served area of legal assistance through intelligent systems. The proposed method’s capability to enhance legal consultation access, particularly where resources are limited, is impactful. The combination of agents to address planning, memory, and adaptability has the potential to set a foundation for future developments in legal multi-agent systems.

Pros
1.	Innovative Approach: Utilizes GLM-4 and multi-agent collaboration, which are advanced concepts in AI, to tackle challenges in the legal domain.
2.	Clear System Design: The roles of Processor, Memorizer, Actor, Judge, and Reflector agents are well-defined, each contributing uniquely to the overall goal.
3.	Addresses Real-World Needs: The system could significantly impact regions with limited legal resources by improving access to legal services.

Cons
1.	Limited Practical Examples: The absence of detailed examples of real-world scenarios could make it challenging for readers to fully grasp the application potential.
2.	Clarity in Technical Details: Some technical descriptions are overly abstract, which may hinder understanding for readers less familiar with multi-agent frameworks.
3.	Scalability Concerns: Although the paper describes a promising model, the scalability of GoalAct in larger, more complex legal systems remains unexplored.

---

### Official Review · ~Yufei_Zhuang1 · 2024-11-09
**Good idea but need more specific details needed**

**Rating:** 7
**Confidence:** 4

**Review:**

On one hand, the detailed description of the functions of each agent, including the Processor, Memorizer, Actor, Judge, and Reflector, and the overall collaborative process framework is well - structured. It theoretically elaborates how the system should operate to handle user requests and solve problems. For instance, it clearly states the tasks of different agents in each stage. This shows a systematic design concept.
However, on the other hand, there is a lack of more specific implementation details.  When it comes to the GoalAct strategy for balancing local and global aspects, although it describes what to do in each dimension, details such as how to precisely adjust the global plan based on dynamic feedback and how to quantitatively judge whether the overall objective is met during reflection are not provided.

---

### Official Review · ~Rosalie_Butte1 · 2024-11-09
**Review of “GoalAct: A Globally Adaptive Dynamic Legal Multi-agent Collaboration System”**

**Rating:** 8
**Confidence:** 4

**Review:**

The paper proposes a method to leverage a multi-agent collaboration system to handle complex legal inquiries. The use of multiple type of agents, helps to break down the problem and enhance the accuracy of the solution by using memory and feedback. To further enhance the quality of the generated solution, the paper proposes to balance the local accuracy of the agents with the global alignment to the overall objective.

The paper targets an important real-world problem to improve accessibility of legal services. It shows a well-structured approach by defining the set of agents and their respective tasks, as well as how these agents work together to solve the overall objective. It also combines an innovative approach to use feedback and a mix of global and local accuracies to further enhance the quality and correctness of the generated solution, which are necessary in the field of legal services.

However, the paper is a bit vague in the description of the technical details, for example how to manage global and local accuracy and how to adjust the global goal. Additionally, the paper could benefit from formulating a plan on how to evaluate and compare the results of GoalAct.

---

### Official Review · ~Yuanda_Zhang1 · 2024-11-09
**Innovative Multi-agent System**

**Rating:** 8
**Confidence:** 4

**Review:**

The proposal introduces GoalAct, a novel dynamic legal multi-agent collaboration system that integrates global and local strategies to provide efficient legal services. The system leverages the GLM4 language model and API calls to legal databases, aiming to enhance accuracy and adaptability in legal assistance.

Pros:
1)The integration of global and local planning, reflection, and memory in the multi-agent system is a sophisticated approach to addressing the complexity of legal inquiries.
2)The use of API calls to access legal databases suggests a practical application that can significantly improve the accessibility and cost-effectiveness of legal services.
3)The system's design to filter irrelevant information and avoid local search loops demonstrates an understanding of the challenges in legal AI and proposes valid solutions.

Cons:
1)The proposal could benefit from a more detailed explanation of how the system handles the diversity of legal issues and the types of errors it can correct in user queries.
2)The balance between local task accuracy and global objective alignment is mentioned as a challenge, but the proposal does not provide a clear strategy for achieving this balance.

---

### Official Review · ~André_Moreira_Leal_Leonor1 · 2024-11-09
**Strengths in Multi-Agent Design with Room for Technical Depth and Risk Considerations**

**Rating:** 8
**Confidence:** 4

**Review:**

The proposal presents a clear, well-structured vision of the GoalAct legal multi-agent system based on GLM4, for improving legal services through adaptive collaboration. One of the strengths of this work is its clear architecture with differentiated agent roles and an innovative combination of global and local planning, reflection, and memory. This multi-agent approach will enhance the accuracy of responses and adaptability, establishing the system as quite useful in the area of legal service automation.

The proposal could be improved by providing more detailed technical descriptions—especially on error-handling mechanisms and memory utilization—to be applicable legally. Further, addressing potential risks with respect to ethical concerns in legal decision-making and API limitations would make the proposal much stronger.

---

### Official Review · ~Chengming_Shi1 · 2024-11-11

**Rating:** 8
**Confidence:** 4

**Review:**

### Summary

The proposal “GoalAct: A Globally Adaptive Dynamic Legal Multi-agent Collaboration System” introduces a sophisticated multi-agent system designed to provide legal services using the GLM-4 language model. The system integrates global and local information processing, planning, reflection, and memory to enhance accuracy and adaptability in legal consultations. The goal is to improve the efficiency and accessibility of legal services through advanced AI technology.

### Pros

1. **Integration of LLMs**: Leveraging Large Language Models like GLM-4 for legal services is innovative and has the potential to revolutionize the legal industry.
2. **Multi-Agent Collaboration**: The use of different types of agents (Processor, Memorizer, Actor, Judge, and Reflector) can effectively decompose complex legal tasks into manageable subtasks.
3. **Adaptability**: The system’s ability to dynamically adjust plans and learn from interactions can lead to more accurate and reliable legal.
4. **Memory Functionality**: Incorporating short-term and long-term memory can enhance the system’s problem-solving capabilities and improve over time.
5. **Global and Local Integration**: The approach of balancing local accuracy with global objectives is a novel strategy that could lead to more effective multi-agent systems.

### Cons

1. **Complexity**: The system’s complexity may make it difficult to implement and maintain, potentially leading to high costs and technical challenges.
2. **Ethical and Legal Considerations**: The use of AI in legal services raises ethical concerns about the quality of advice and the potential for malpractice.
3. **Dependence on Data Quality**: The system’s performance is highly dependent on the quality and relevance of the legal databases it accesses.
4. **User Trust**: Building trust in AI-driven legal services may be challenging, especially given the sensitive nature of legal matters.
5. **Scalability**: The system may face scalability issues, as it needs to be adaptable to a wide range of legal issues and jurisdictions.

---

### Official Review · ~jin_wang30 · 2024-11-12
**Good proposal**

**Rating:** 9
**Confidence:** 4

**Review:**

This article provides an innovative legal multi-agent collaboration system, which has theoretical research value.

Advantages：
The article adopts a standard academic structure, including background introduction, definition, related work and methods, with clear levels and easy for readers to understand.
In addition, the article cites a large number of recent studies in the field of LLM, such as Chain-of-Thought, ReAct, Reflexion and other methods, showing the author's understanding and use of cutting-edge technologies in the field.
As for the specific implementation of the GoalAct system, the role and process of each agent (Processor, Memorizer, Actor, Judge, Reflector) are clearly described, reflecting the logical rigor of the system.

Disadvantages：
The innovation of the article does not seem to be enough. The article is mainly based on GLM-4 and calls the relevant database API interface, but there is no innovation in the improvement of the model itself. In addition, this article lacks a discussion of potential challenges. The multi-agent collaboration system may have challenges in scalability and real-time performance, but the article does not explore these potential problems and solutions in depth.

---

### Official Review · ~Chaoqun_Yang2 · 2024-11-12
**A practical application**

**Rating:** 8
**Confidence:** 4

**Review:**

**Summary:**
The paper introduces a novel approach to legal service provision through a multi-agent system that leverages the capabilities of Large Language Models (LLMs). The system, named GoalAct, is designed to address the complexities of legal inquiries by integrating global and local information, utilizing APIs to access legal databases. The proposed system consists of five types of agents: Processor, Memorizer, Actor, Judge, and Reflector, each with specific roles in processing user demands and providing solutions.

**Highlights:**
1. **Practical Application:** The paper emphasizes the practical implications of the system, suggesting that it could significantly impact the legal industry by making services more efficient and accessible.
2. **Comprehensive System Design:** The system's design, which includes five distinct types of agents, is comprehensive and addresses multiple facets of legal service provision, from processing and memory to action planning and reflection.

**Advice:**
1. **Related Work Depth:** The review of related work is somewhat brief. Expanding on this section to include a more detailed analysis of existing systems and their limitations would provide a stronger context for the proposed method. Additionally, discussing how GoalAct overcomes these limitations would be valuable.
4. **Innovative Methodology:** It would be better if the proposed method can demonstrate differences from existing techniques. For instance, what's the difference between the proposed method and the current mainstream multi-agent collaboration approach?

---

### Official Review · ~Zihan_Wang7 · 2024-11-12
**Reflexion LLM of Law**

**Rating:** 8
**Confidence:** 5

**Review:**

**Summary**

This paper proposes GoalAct, a legal multi-agent collaboration system based on GLM-4 that combines global and local information processing to provide legal services. The system employs five specialized agents (Processor, Memorizer, Actor, Judge, and Reflector) working collaboratively to handle legal queries while maintaining a balance between local task accuracy and global objectives.

**Highlights**

- Reflector: In addition to conventional memory and execution agents, the system incorporates a reflection module to optimize and improve solutions
- Technical Foundation: Built upon recent advances such as CoT, ReAct, and Reflexion frameworks

**Advice**

The paper could benefit from more specific details:

- Lacks investigation into legal database APIs
- Missing specific mechanisms for legal-domain reflection

---

### Official Review · ~Zhu_Zhang6 · 2024-11-12
**A good proposal for legal multi-agent system**

**Rating:** 7
**Confidence:** 4

**Review:**

**Summary:**

The proposal outlines "GoalAct," a globally adaptive dynamic legal multi-agent collaboration system based on the GLM-4 language model. The system is designed to provide efficient and accurate legal services by integrating global and local information. By using API calls to access legal databases, GoalAct addresses legal service challenges, such as handling complex inquiries, managing irrelevant information, and developing a robust self-correction mechanism. The system employs five types of agents—Processor, Memorizer, Actor, Judge, and Reflector—that collaboratively tackle user demands by breaking down and processing tasks.

**Strengths:**

1. **Innovative Agent Design:** The structured multi-agent system, with distinct agents for processing, memory management, action, judgment, and reflection, allows for a highly modular approach that is potentially scalable and adaptable to complex legal inquiries.
2. **Memory Utilization:** The integration of both short-term and long-term memory is particularly promising for improving decision accuracy over time and could reduce error recurrence.

**Weaknesses:**

1. **Evaluation Metrics Not Defined:** There is minimal information on how the system’s success or effectiveness would be measured, making it difficult to assess its real-world impact.

**Questions:**

1. How will GoalAct handle potential conflicts or redundancies between agents, especially in complex or ambiguous cases?
2. What specific metrics or benchmarks will be used to evaluate the system’s effectiveness, particularly for different legal domains?

---

### Official Review · ~Chendong_Xiang1 · 2024-11-12

**Rating:** 7
**Confidence:** 3

**Review:**

The paper presents GoalAct, a globally adaptive multi-agent system leveraging the GLM-4 language model for dynamic legal service provision.
Strengths
1.Adaptive Collaboration: GoalAct’s multi-agent system enables efficient collaboration by assigning specialized tasks to different agents (Processor, Memorizer, Actor, Judge, Reflector), which enhances both task accuracy and overall system adaptability.
2.Memory Integration: By utilizing both short- and long-term memory, the system accumulates past experiences and improves decision-making over time, contributing to more robust and contextually accurate legal services.
3.Dynamic Reflection and Adjustment: GoalAct continuously reflects on planning and execution, dynamically adjusting strategies based on user feedback, which enhances reliability and responsiveness.
questions：
dose authors show ablation of system design that claim of Processor, Memorizer, Actor, Judge, Reflector act critically?